# ADAPTIVE WEIGHT DECAY: ON THE FLY WEIGHT DECAY TUNING FOR IMPROVING ROBUSTNESS

## ABSTRACT

We introduce adaptive weight decay, which automatically tunes the hyper-parameter for weight decay during each training iteration. For classification problems, we propose changing the value of the weight decay hyper-parameter on the fly based on the strength of updates from the classification loss (i.e., gradient of cross-entropy), and the regularization loss (i.e., $\ell_2$-norm of the weights). We show that this simple modification can result in large improvements in adversarial robustness — an area which suffers from robust overfitting — without requiring extra data. Specifically, our reformulation results in 20% relative robustness improvement for CIFAR-100, and 10% relative robustness improvement on CIFAR-10 comparing to traditional weight decay. In addition, this method has other desirable properties, such as less sensitivity to learning rate, and smaller weight norms, which the latter contributes to robustness to overfitting to label noise, and pruning.

## 1 INTRODUCTION

Modern deep learning models have exceeded human capability on many computer vision tasks. Due to their high capacity for memorizing training examples (Zhang et al., 2021), their generalization heavily relies on the training algorithm. To reduce memorization, several approaches have been taken including regularization and augmentation. Some of these augmentation techniques alter the network input (DeVries & Taylor, 2017; Chen et al., 2020; Cubuk et al., 2019; 2020; Müller & Hutter, 2021), some alter hidden states of the network (Srivastava et al., 2014; Ioffe & Szegedy, 2015; Gastaldi, 2017; Yamada et al., 2019), some alter the expected output (Warde-Farley & Goodfellow, 2016; Kannan et al., 2018), and some effect multiple levels (Zhang et al., 2017; Yun et al., 2019; Hendrycks et al., 2019b). Another popular approach to prevent overfitting is the use of regularizers, such as weight decay (Plaut et al., 1986; Krogh & Hertz, 1991). Such methods prevent over-fitting by eliminating solutions that memorize training examples. Regularization methods are attractive beyond generalization on clean data as they are crucial in adversarial and noisy-data settings. In this paper, we revisit weight decay; a regularizer mainly used to avoid overfitting.

The rest of the paper is organized as follows: In Section 2, we revisit tuning the hyper-parameters for weight decay. We introduce adaptive weight decay in Section 3, and further discuss its properties in 4. More specifically, we discuss the benefits of adaptive weight decay in the setting of adversarial training in subsection 4.1, noisy labels in subsection 4.2, and additional properties related to robustness in subsection 4.3.

## 2 WEIGHT DECAY

Weight decay which encourages weights of networks to have smaller magnitudes (Zhang et al., 2018) has widely been adopted to improve generalization. Although other forms of weight decay have been studied (Loshchilov & Hutter, 2017), we focus on the popular $\ell_2$-norm variant. More precisely, we look at classification problems with cross-entropy as the main loss and weight decay as the regularizer, which was popularized by Krizhevsky et al. (2017):

$$Loss_w(x,y) = CrossEntropy(w(x), y) + \frac{\lambda_{wd}}{2}\|w\|_2^2, \qquad (1)$$

|  | Setting 1 | Setting 2 |
|---|---|---|
| $lr$ | 0.01 | 0.1 |
| $\lambda_{wd}$ | 0.005 | 0.0005 |
| Weight Decay Update $(-w\lambda_{wd}lr)$ | $-w_t \times 5 \times 10^{-5}$ | $-w_t \times 5 \times 10^{-5}$ |
| Cross Entropy Update $(-\nabla w_t lr)$ | $-\nabla w_t \times \mathbf{10^{-2}}$ | $-\nabla w_t \times \mathbf{10^{-1}}$ |

Table 1: Effect of moving along the diagonal on the 2D grid search for learning rate and $\lambda_{wd}$ on the updates in one gradient descent step.

where $w$ is the network parameters, and $(x, y)$ is the training data, and $\lambda_{wd}$ is the hyper-parameter controlling how much weight decay penalizes the norm of weights compared to the main loss (i.e., cross-entropy loss). For instance, if $\lambda_{wd}$ is negligible, the optimization will likely over-fit data, whereas if $\lambda_{wd}$ is too large, the optimization will collapse to a low-weighted solution that does not fit the training data. Consequently, finding the correct value for the weight decay's hyper-parameter is crucial. Models trained using the right hyper-parameter for weight decay tend to have higher bias, which translates to less over-fitting and, thus, better generalization (Krogh & Hertz, 1991).

## 2.1 TUNING HYPER-PARAMETERS

When tuning $\lambda_{wd}$, it is crucial to search for learning-rate $lr$ simultaneously, [1] as illustrated in Figure 1 for CIFAR-10 and CIFAR-100. Otherwise, in the case that the 2D grid search is not computationally practical, we show that separate 1D grid search on learning rate and weight decay is not optimal. See Appendix C.

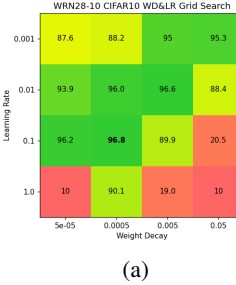
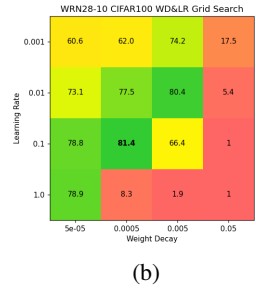
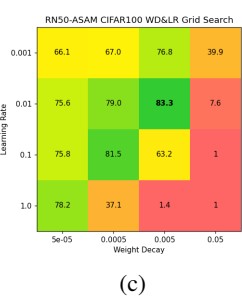

(a)  (b)  (c)

Figure 1: Grid Search on different values of learning rate and weight decay on accuracy of WRN28-10 trained with SGD on (a) CIFAR10 and (b) CIFAR100, and (c) ResNet50 optimized with ASAM (Kwon et al., 2021) for CIFAR-100.

To better understand the relationship between adjacent cells on the same diagonal of the grid search, let us consider the two cells corresponding to sets of hyper-parameter values ($\lambda_{wd} = 0.005, lr = 0.01$) and ($\lambda_{wd} = 0.0005, lr = 0.1$). We compare one step of gradient descent using these hyper-parameters. To derive the parameter $w$ at step $t + 1$ from its value at step $t$ we have:

$$w_{t+1} = w_t - \nabla w_t lr - w\lambda_{wd}lr, \tag{2}$$

where $\nabla w_t$ is the gradient computed from the cross-entropy loss and $w\lambda_{wd}$ is the gradient computed from the weight decay term from eq. 1. By comparing the two settings, we realize that the update coming from weight decay (i.e. $-w\lambda_{wd}lr$) remains the same, but the update coming from cross entropy (i.e. $-\nabla w_t lr$) differs by a factor of 10 as shown in Table 1. In other words, by moving along cells on the same diagonal in Fig. 1, we are changing the importance of the cross-entropy term compared to weight decay while keeping the updates from the weight decay intact. A question we ask is: *What is the optimal ratio between the update coming from cross-entropy and the update coming from the weight decay in settings that generally perform better?*

---

[1]See Appendix C for more details on the importance of 2D grid search.

To answer, we study the value of weight **D**ecay **o**ver **G**radient of cross-entropy (DoG for short) as a metric:

$$DoG_t = \frac{\|\lambda_{wd} w_t\|}{\|\nabla w_t\|},$$

(3)

More specifically, we are interested to track the ratio of $DoG$ from eq. 3 at every iteration for settings that achieve higher validation accuracy. Hence, we focus on CIFAR-100 and on setting 2 of Table 1. In particular, we plot the loss of the cross-entropy loss –in logarithmic scale– for each epoch in Fig. 2(b), and plot the running average of the $DoG$ values for all iterations in Fig. 2(a). For this experiment, the loss plateaus at epoch 350 (Fig. 2(b)), for which the running average of $DoG$ at that epoch is $DoG = 0.016$. This average value of $DoG = 0.016$ gives a rough idea of the ratio between the two updates over the course of training.

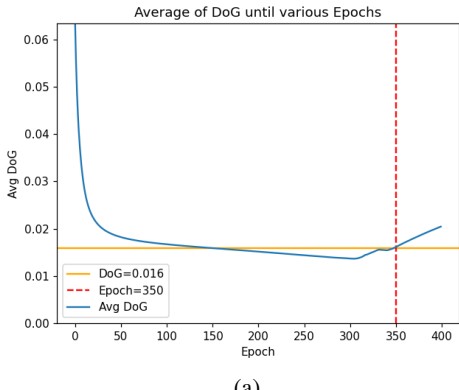
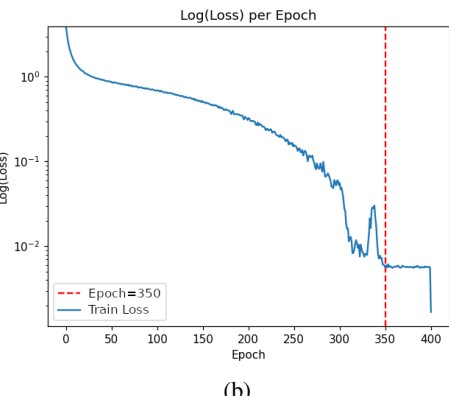

(a)                                                                 (b)

Figure 2: How to find the $DoG$ value associated with a fixed $\lambda_{wd}$ hyper-parameter. The average $DoG = 0.016$ is computed by averaging the $DoG$ values from all iterations from the beginning of the training until epoch 350 (a), where 350 is the epoch at which the training loss has plateaued (b).

If the ratio $DoG = 0.016$ is the underlying difference causing different performances in different cells in the same diagonal, ensuring this value is constantly seen in every optimization step might have great benefits.

## 3 ADAPTIVE WEIGHT DECAY

At this point, we introduce another simple yet effective way of maintaining this golden ratio $DoG = 0.016$ that we call Adative Weight Decay. Adaptive weight decay is similar to non-adaptive (i.e., traditional) weight decay. The only difference is that the hyper-parameter $\lambda_{wd}$ changes in every iteration to ensure that $DoG = 0.016$ is always satisfied. To keep this constant ratio at every step $t$, we can rewrite the $DoG$ equation (eq. 3) as:

$$\lambda_t = \frac{0.016 \cdot \|\nabla w_t\|}{\|w_t\|},$$

(4)

Eq 4 allows us to have a different weight decay hyper parameter value ($\lambda_{wd}$) for every optimization iteration $t$, which keeps the gradients received from the cross entropy and weight decay balanced throughout the optimization. Note that this value for weight decay penalty $\lambda_t$ can be computed on the fly with almost no computational overhead during the training.

In Table 2, we compare networks trained with non-adaptive weight decay using the optimal $lr$ and $\lambda_{wd}$ from Fig. 1 to those trained with adaptive weight decay where we estimate the $DoG$ value similar to what was mentioned before in Fig. 2.

We can see that networks trained with adaptive weight decay show comparable performance to those trained with non-adaptive weight decay in terms of held-out validation accuracy and also test

| Dataset | $lr$ | $\lambda_{wd}$ | $DoG$ | Val Acc | Test Acc | $\|W\|_2$ | $\frac{0.0005 \cdot \|W\|_2^2}{2}$ | X-ent | Total Loss |
|---------|------|---------|-------|---------|----------|-----------|------------------|-------|------------|
| C100 | 0.1 | 5e-4 | - | 80.97 | 80.53 | 24.3 | 0.15 | 0.00 | 0.15 |
|      | 0.1 | - | 0.016 | 80.91 | 80.75 | **18.3** | 0.08 | 0.01 | **0.09** |
| C10 | 0.1 | 5e-4 | - | 96.83 | 96.31 | 17.2 | 0.07 | 0.00 | 0.08 |
|     | 0.1 | - | 0.018 | 96.76 | 96.00 | **8.4** | 0.02 | 0.02 | **0.03** |

Table 2: WRN28-10 Models trained with adaptive weight decay by estimating the DoG value from a model trained with non-adaptive weight decay. Training with adaptive weight decay result in models with smaller weight-norms and even smaller total training loss. Results are an average of 3 runs and X-ent is the average training cross-entropy at the end of training. Dataset names are abbreviated, C100 is CIFAR-100.

accuracy. However, networks trained with adaptive weight decay have considerably smaller weight norms. Most interestingly, when we measure the non-adaptive total loss (from eq. 1), we observe that networks trained with the adaptive method have smaller training loss even though they have not optimized that loss directly. This implies that adaptive weight decay can find solutions with smaller objective function value than those found by non-adaptive weight decay.

To study the differences between adaptive and non-adaptive weight decay, we can plug in $\lambda_t$ of the adaptive method (eq. 4) directly into the equation for traditional weight decay (eq. 1) and derive at the total loss based on Adaptive Weight Decay:

$$Loss_{w_t}(x,y) = CrossEntropy(w_t(x), y) + 0.008\|\nabla w_t\|\|w_t\|, \qquad (5)$$

where $\|\nabla w_t\|$ in eq. 5 is treated as a constant and we do not allow gradients to flow through it based on the definition of adaptive weight decay. In general, adaptive weight decay has negligible computation overhead compared to training with traditional non-adaptive weight decay. The implementation details and pseudo-code for our algorithm can be found in Appendix B. By comparing the weight decay term in the adaptive weight decay's final loss (eq. 5): $\frac{DoG}{2}\|w\|\|\nabla w\|$ with that of the traditional weight decay final loss (eq. 1): $\frac{\lambda_{wd}}{2}\|w\|^2$, we can build intuition on some of the differences between the two. For example, the non-adaptive weight decay regularization term approaches zero only when the weight norms are close to zero, whereas, in adaptive weight decay, it also happens when the cross-entropy gradients are close to zero. This is a desirable property since, intuitively, when the optimizer finds a flat minima, adaptive weight decay stops over-optimizing the norm of weights, whereas, in non-adaptive weight decay, that is not the case. Also, when the gradient of cross-entropy is large, adaptive weight decay penalizes the norm of the weights more which could prevent the optimization from falling into a steep local minima which could result in overfitting.

## 4 PROPERTIES OF ADAPTIVE WEIGHT DECAY

In this section, we discuss the properties of adaptive weight decay compared to its non-adaptive counterpart. First, we will look into adversarial robustness and noisy data applications – two areas which could benefit the most from the potentials of adaptive weight decay. For adversarial robustness, we will look at the robust overfitting phenomenon (Rice et al., 2020) and study the effect of adaptive weight decay on reducing robust overfitting. We observe the relative performance improvement of 10.71% on CIFAR-10 and 20.51% on CIFAR-100 (i.e. 4% absolute improvement in robustness for CIFAR-100 and 1% on CIFAR-10) over the popular baseline of 7-step PGD adversarial training (Madry et al., 2017). Then, we study the effect of adaptive weight decay in the noisy label setting. More specifically, we show roughly 4% improvement in accuracy on CIFAR-100 and 2% on CIFAR-10 for training on 20% symmetry label flipping setting (Bartlett et al., 2006). Then, we shift our focus to sub-optimal learning rates. We show that adaptive weight decay finds better optima when the learning rate is not tuned. We also show that networks achieving roughly similar accuracy, once trained with adaptive weight decay, tend to have lower weight norms. This phenomenon can have exciting implications for pruning networks (LeCun et al., 1989; Hassibi & Stork, 1992).

### 4.1 Adversarial Robustness

Deep neural networks are susceptible to adversarial perturbations (Szegedy et al., 2013; Biggio et al., 2013). In the adversarial setting, the adversary adds a small imperceptible noise to the image which fools the network to make an incorrect prediction. To ensure that the adversarial noise is imperceptible to the human eye, usually noise with $\ell_p$-norm bounds have been studied (Sharif et al., 2018). In such settings, the objective for the adversary is to maximize the following loss:

$$\max_{|\delta|_p \leq \epsilon} CrossEntropy(w(x + \delta), y),$$ (6)

An array of works study methods of generating strong adversarial examples (Goodfellow et al., 2014; Madry et al., 2017; Carlini & Wagner, 2017; Izmailov et al., 2018; Croce & Hein, 2020a; Andriushchenko et al., 2020). While some studies focus on defenses with theoretical gaurantees (Wong & Kolter, 2018; Cohen et al., 2019), variations of adversarial training have been the de-facto defense against adversarial attacks (Madry et al., 2017; Shafahi et al., 2019; Wong et al., 2020; Rebuffi et al., 2021; Gowal et al., 2020).

Adversarial training consists of generating adversarial examples for the training data on the fly and training on them. More precisely, the loss for adversarial training is as follows:

$$\min_{w} \big( \max_{|\delta|_p \leq \epsilon} CrossEntropy(w(x + \delta), y) + \frac{\lambda_{wd}}{2} \|w\|_2^2 \big),$$ (7)

While adversarial training is a strong baseline for defenses against adversarial attacks, it usually suffers from a phenomenon called Robust Overfitting (Rice et al., 2020). To overcome this issue, various methods have been proposed to rectify robust overfitting, including early stopping (Rice et al., 2020), use of extra unlabeled data (Carmon et al., 2019), synthesized images (Gowal et al., 2020), pre-training (Hendrycks et al., 2019a), use of data augmentations (Rebuffi et al., 2021), and stochastic weight averaging (Izmailov et al., 2018). We re-visit tuning conventional non-adaptive weight decay parameter ($\lambda_{wd}$) for the task of improving robustness generalization, and compare that to tuning the $DoG$ hyper-parameter for adaptive weight decay.

For these experiments, we use a WideResNet 28-10 architecture and use widely accepted PGD adversarial training to solve eq. 7 (Madry et al., 2017) while keeping 10% of the examples from the training set as held-out validation set for the purpose of early stopping. For early-stopping, we select the checkpoint which gives the highest $\ell_\infty = 8$ robustness accuracy measured by a 20 step PGD attack on the held-out validation set. Other details about the experimental setup can be found in Appendix A.1.

To ensure that we search for enough values for $\lambda_{wd}$, we use twice as many values for $\lambda_{wd}$ compared to $DoG$. Figure 3 plots the robustness accuracy measured by applying AutoAttack (Croce & Hein, 2020b) on the test examples for the CIFAR-10 and CIFAR-100 datasets, respectively. We observe that training with adaptive weight decay improves the robustness by a margin of 4.84% on CIFAR-10 and 5.08% on CIFAR-100, compared to the non-adaptive counterpart. This is a relative improvement of 10.7% and 20.5% on CIFAR-10 and CIFAR-100, respectively.

Increasing robustness often comes at the cost of drops in clean accuracy (Zhang et al., 2019). This could be partially due to the fact that some $\ell_p$-norm bounded adversarial examples look more like the network's prediction than their original class(Sharif et al., 2018). An active area of research seeks a better trade-off between robustness and natural accuracy by finding other points on the pareto-optimal curve of robustness and accuracy. For example, (Balaji et al., 2019) use instance-specific perturbation budgets during training. Interestingly, when comparing the most robust network trained with non-adaptive weight decay ($\lambda_{wd}$) to that trained with adaptive weight decay ($DoG$), we notice that those trained with adaptive weight decay have higher clean accuracy (Table. 3). In addition, similar to the case before, we can observe comparatively smaller weight-norms for models which are adversarially trained with adaptive weight decay which might contribute to their better generalization.

The previously shown results suggest excellent potential for adversarial training with adaptive weight decay. To further study this potential, we only substituted the Momentum-SGD optimizer with the ASAM optimizer (Kwon et al., 2021) and used the same hyperparameters used in previous

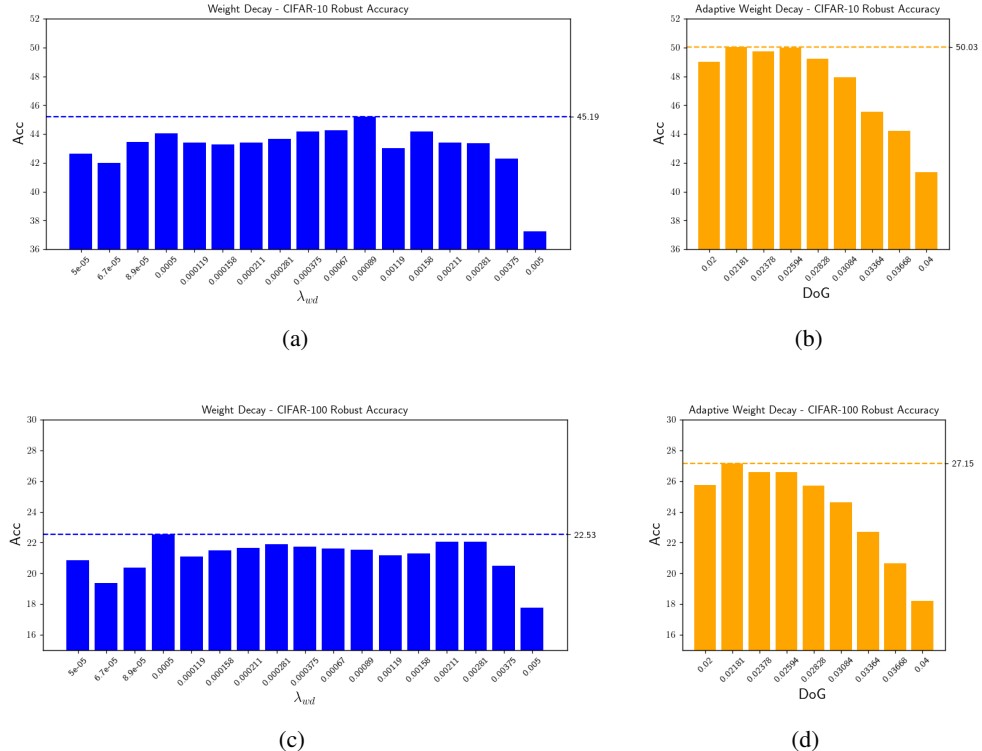

Figure 3: $\ell_\infty = 8$ robust accuracy on the test set of adversarially trained WideResNet28-10 networks on CIFAR-10 and CIFAR-100. (a, c) Using different choices for the hyper-parameter of non-adaptive weight decay ($\lambda_{wd}$), and (b, d) different choices of the hyper-parameter for adaptive weight decay ($DoG$).

| Method | Dataset | Optimizer | $\|W\|_2$ | Nat Acc | AutoAtttack |
|---|---|---|---|---|---|
| $\lambda_{wd} = 0.00089$ | CIFAR-10 | SGD | 35.58 | 84.31 | 45.19 |
| $DoG = 0.022$ | | SGD | **7.11** | **87.08** | **50.03** |
| $\lambda_{wd} = 0.0005$ | CIFAR-100 | SGD | 51.32 | 60.15 | 22.53 |
| $DoG = 0.022$ | | SGD | **13.41** | **61.39** | **27.15** |

Table 3: Adversarial robustness of WRN28-10 PGD-7 adversarially trained networks using adaptive and non-adaptive weight decay. Table summarizes the best performing hyper-parameter for each method on each dataset. Not only adaptive weight decay outperforms the non-adaptive weight decay trained model in terms of robust accuracy, it is also superior in terms of the natural accuracy.

experiments. To the best of our knowledge, and according to the RobustBench benchmark (Croce et al., 2020), the state-of-the-art $\ell_\infty = 8.0$ robust accuracy for CIFAR-100 without extra synthesized or captured data using WRN28-10 is 29.80% (Rebuffi et al., 2021). We achieve 29.54% robust accuracy, which is comparable to this baseline. See Table 4 for more details.

In Appendix. E.5 we provide more experiments about training robust models for larger datasets and for cases where robust overfitting is not an issue. On ImageNet, we illustrate that robust models trained with adaptive-weight decay have better overall performance compared to those trained with their non-adaptive weight decay.

## 4.2 ROBUSTNESS TO NOISY LABELS

Popular vision datasets, such as MNIST (LeCun & Cortes, 2010), CIFAR (Krizhevsky et al., 2009), and ImageNet (Deng et al., 2009), contain some amount of label noise (Yun et al., 2021; Zhang, 2017). While some studies provide methods for identifying and correcting such errors (Yun et al., 2021; Müller & Markert, 2019; Al-Rawi & Karatzas, 2018; Kuriyama, 2020), others provide training

| Method | WRN | Aug | Epo | ASAM | TR | SWA | Nat | AA |
|---|---|---|---|---|---|---|---|---|
| $\lambda_{wd} = 0.0005$ | 28-10 | P&C | 200 | - | - | - | 60.15 | 22.53 |
| $\lambda_{wd} = 0.0005$ | 28-10 | P&C | 100 | ✓ | - | - | 58.09 | 22.55 |
| $\lambda_{wd} = 0.00281*$ | 28-10 | P&C | 100 | ✓ | - | - | 62.24 | 26.38 |
| $\lambda_{AdaDecay} = 0.002*$ | 28-10 | P&C | 200 | - | - | - | 57.17 | 24.18 |
| $DoG = 0.022$ | 28-10 | P&C | 200 | - | - | - | 61.39 | 27.15 |
| $DoG = 0.022$ | 28-10 | P&C | 100 | ✓ | - | - | **63.93** | 29.54 |
| (Rebuffi et al., 2021) | 28-10 | CutMix | 400 | - | ✓ | ✓ | 62.97 | **29.80** |
| (Rebuffi et al., 2021) | 28-10 | P&C | 400 | - | ✓ | ✓ | 59.06 | 28.75 |

Table 4: CIFAR-100 adversarial robustness performance of various strong methods. Adaptive weight decay with ASAM optimizer outperforms many strong baselines. For experiments marked with * we do another round of hyper-parameter search. $\lambda_{AdaDecay}$ indicates using the work from Nakamura & Hong (2019). The columns represent the method, depth and width of the WideResNets used, augmentation, number of epochs, whether ASAM, TRADES (Zhang et al., 2019), and Stochastic Weight Averaging (Izmailov et al., 2018), were used in the training, followed by the natural accuracy and adversarial accuracy using AutoAttachk. In the augmentation column, P&C is short for Pad and Crop. Note that both $DoG$ and $\lambda_{wd}$ are simply trained using 7step PGD adversarial training.

algorithms to avoid over-fitting noisy training data or, even better, avoid fitting the noisy labeled examples entirely (Jiang et al., 2018; Song et al., 2019; Jiang et al., 2020).

In this section, we perform a preliminary investigation of adaptive weight decay's resistance to fitting data with label noise. Following previous studies, we use symmetry label flipping (Bartlett et al., 2006) to create noisy data for CIFAR-10 and CIFAR-100 and use ResNet34 as the backbone. Other experimental setup details can be found in Appendix A.2. Similar to the previous section, we test different hyper-parameters for adaptive and non-adaptive weight decay. To ease comparison in this setting, for each hyper-parameter, we train two networks: 1- with a certain degree of label noise, 2- with no noisy labels. We then report the accuracy on the test set where the test labels are not altered. The test accuracy on the second network – one which is trained with no label noise – is just the clean accuracy. Having the clean accuracy coupled with the accuracy after training on noisy data enables an easier understanding of the sensitivity of each training algorithm and choice of hyper-parameter to label noise. Figure 4 gathers the results for the noisy data experiment on CIFAR-100.

Figure 4 illustrates that networks trained with adaptive weight decay have a smaller drop in performance when there is label noise in the training set in comparison to non adaptive weight decay. For instance, Figure 4(a) shows that $DoG = 0.028$ for adaptive and $\lambda_{wd} = 0.0089$ for non-adaptive weight decay achieve roughly 70% accuracy when trained on clean data, while the adaptive version achieves 4% higher accuracy when trained on the 20% label noise setting compared to the non-adaptive version. Similar conclusions can be drawn for CIFAR-10 for which the experiments can be found in Appendix E.1.

Intuitively, from the adaptive weight decay equation in eq. 4, towards the end of training, where the examples with label noise are producing large gradients adaptive weight decay increases the penalty for weight decay which prevents fitting the noisy data by regularizing the gradient.

## 4.3 ADDITIONAL ROBUSTNESS BENEFITS

Throughout the 2D grid search experiments in section 2, we observed that non-adaptive weight decay is sensitive to changes in the learning rate (LR). In this section we aim to study the sensitivity of the best hyper-parameter value for adaptive and non-adaptive weight decay to learning rate. In addition, models trained with adaptive weight decay tend to have smaller weight norms which could make them more suited for pruning. To test this intuition, we adopt a simple non-iterative $\ell_1$ pruning. To build confidence on robustness to LR and pruning, for the optimal choices of $\lambda_{wd} = 0.0005$ and the estimated $DoG = 0.016$ for WRN28-10, we train 5 networks per choice of learning rate. We prune each network to various degrees of sparsity. We then plot the average of all trials per parameter set for each of the methods. Fig. 5 summarizes the clean accuracy without any pruning and the accuracy after 70% of the network is pruned. As it can be seen, for CIFAR-100, adaptive weight decay is both

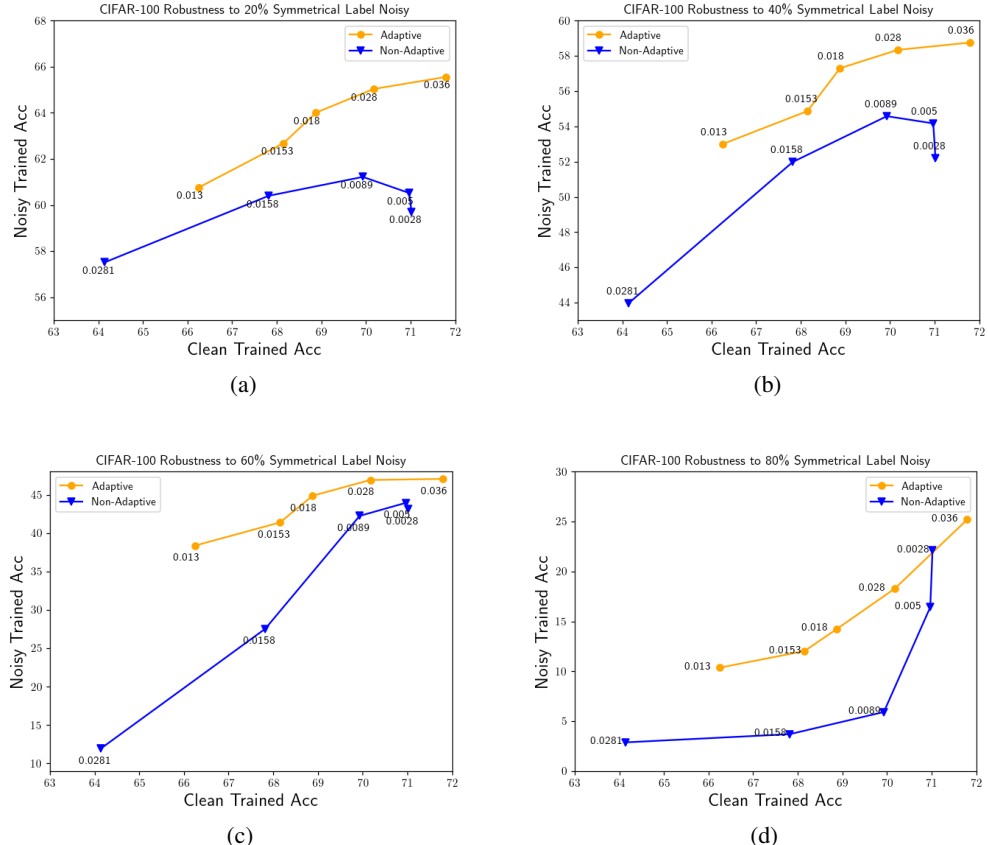

Figure 4: Comparison of similarly performing networks once trained on CIFAR-100 clean data, after training on 20% (a), 40% (b), 60% (c), and 80% (d) noisy data. Networks trained with adaptive weight decay are less sensitive to label noise compared to ones trained with non-adaptive weight decay.

more robust to learning rate changes and also the result networks are less sensitive to parameter pruning. For more details and results on CIFAR-10, please see Appendix E.4 and E.6.

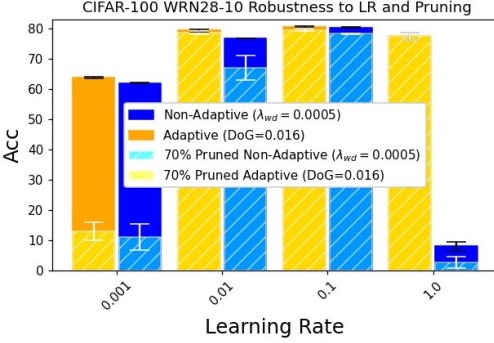

Figure 5: CIFAR-100 models trained with Adaptive Weight Decay (AWD) are less sensitive to learning rate. Also, due to the smaller weight norms of models trained with AWD, they seem like good candidates for pruning. Interestingly, when models are trained with smaller learning rates, they could be more sensitive to trivial pruning algorithms such as non-iterative (i.e., global) $\ell_1$ pruning. The results are average of 4 runs.

## 4.4 UNDER-FITTING DATA, A DESIRABLE PROPERTY

Our experiments show that adaptive weight decay prevents fitting all the data in case of noisy label and adversarial training. Experimentally, we showed that adaptive weight decay contributes to this outcome more than non-adaptive weight decay. Interestingly, even in the case of natural training of even simple datasets such as CIFAR-100, networks trained with optimal adaptive weight decay still underfit the data. For instance, consider the following setup where we train a ResNet50, with

ASAM (Kwon et al., 2021) minimizer with both adaptive and non-adaptive weight decay[2]. Table 5 shows the accuracy of the two experiments.

| Method | Learning Rate | $\|W\|_2$ | Test Acc(%) | Training Acc(%) |
|---|---|---|---|---|
| Adaptive $DoG = 0.022$ | 0.01 | 14.39 | 83.21 | 95.29 |
| Non-Adaptive $\lambda_{wd} = 0.005$ | 0.01 | 17.86 | 83.23 | 98.57 |

Table 5: Train and Test accuracy of adaptive and non-adaptive weight decay trained models. While the adaptive version fits 3.28% less training data, it still results in comparable test accuracy. Our hypothesis is that probably it avoids fitting the noisy labeled data.

Intruigingly, without losing any performance, the model trained with adaptive weight decay avoids fitting an extra 3.28% of training data compared to the model trained with non-adaptive weight decay. We investigated more on what the 3.28% unfit data looks like [3]. Previously studies have discovered that some of the examples in the CIFAR-100 dataset have wrong labels (Zhang, 2017; Müller & Markert, 2019; Al-Rawi & Karatzas, 2018; Kuriyama, 2020). We found the 3.28% to be consistent with their findings and that many of the unfit data have noisy labels. We show some apparent noisy labeled examples in Figure 6.

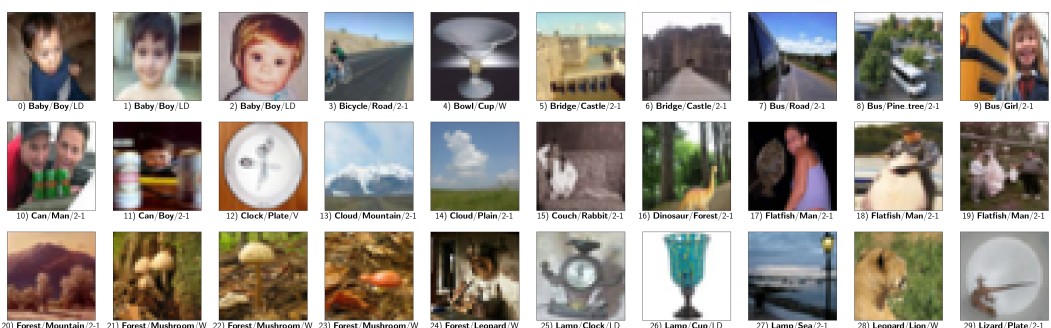

Figure 6: Examples from CIFAR-100 training dataset that have noisy labels. For every image we state [dataset label/ prediction of classifier trained with AWD / our category of noisy case]. We classify these noisy labels into several categories: **W**: Wrong Labels where the picture is clear enough to comprehend the correct label.; **2-1**: Two Objects from CIFAR-100 in one image, but only one label is given in the dataset; **LD**: Loosely Defined Classes where there is one object, but one object could be two classes at the same time. For instance, a baby girl is both a baby and a girl. **V**: Vague images where authors had a hard time identifying.

## 5 CONCLUSION

Regularization methods for a long have aided deep neural networks in genarlizing to data not seen during training. Due to their significant effects on the outcome, it is crucial to have the right amount of regularization and correctly tune training hyper-parameters. We propose Adaptive Weight Decay (AWD) which is a simple modification to one of the more common regularization methods. We compare AWD with non-adaptive weight decay in multiple settings such as adversarial robustness, and training with noisy labels, and experimentally illustrate that AWD results in more robustness.

---

[2]To find the best performing hyper-parameters for both settings, we do a 2D grid-search which can be seen in (c) of Fig. 1.

[3]See Appendix E.3 for images from the 4.71%.

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

## A    EXPERIMENTAL SETUP

In this section, we include the experimental setup used to produce the experiments throughout this paper. We include all hyperparameters used for all experiments, unless explicitly mentioned otherwise. For all experiments, we use SGD optimizers with momentum $\mu = 0.9$. We use a Cosine learning rate schedule with no warm-up and with the value of $0$ for the final value. The weight decay and learning rate for experiments that have not been clearly specified are $\lambda_{wd} = 0.0005$ and $lr = 0.1$. We train all networks for 200 epochs with a batch-size of 128. For all experiments that use ASAM, we use the hyper-parameters the original paper suggests.

### A.1    ADVERSARIAL TRAINING

We use a pre-act WideResNet28 with a width of 10. We use $\ell_\infty = 8/255$ PGD attack with the step size of $2/255$. We use 7 steps for creation of adversarial examples for training and use the minimum accuracy produced by AutoAttack for the test. We keep $10\%$ of training data for validation and use it to do early stopping. For $\lambda_{wd}$ of non-adaptive weight decay, we fit a geometric sequence of length 17 starting from 0.00005 and ending at 0.005. For the adaptive weight decay hyper parameter ($DoG$), we fit a geometric sequence of length 9 starting from 0.02 and ending at 0.04.

### A.2    NOISY LABEL TRAINING

We use a ResNet34 as our architecture. For each setting we used 5 different hyperparameters for adaptive and non-adaptive weight decays. For CIFAR-10, we use $\lambda_{wd} \in \{0.0028, 0.005, 0.0089, 0.0158, 0.0281\}$ and $DoG \in \{0.036, 0.028, 0.018, 0.0153, 0.018\}$ and for CIFAR-100, we use $\lambda_{wd} \in \{0.0281, 9.9158, 0.0089, 0.005, 0.0028\}$ and $DoG \in \{0.013, 0.0153, 0.018, 0.028, 0.036\}$.

## B    IMPLEMENTATION

In this section, we discuss the details of implementation of adaptive weight decay. The method is not really susceptible to the exact implementation details discussed here, however, to be perfectly candid, we include all details here. First, let us assume that we desire to implement adaptive weight decay using $DoG = 0.016$ as the hyperparameter. We know that $\lambda_t = \frac{\|\nabla w_t\| 0.016}{\|w_t\|}$. Please note that $\|\nabla w_t\|$ requires knowing the gradients of the loss w.r.t. the network's parameters. Meaning that to compute the $\lambda_t$ for every step, we have to call a backward pass on the actual parameters of the network. After this step, given the fact that we know both $\|w_t\|$ and $\|\nabla w_t\|$, we can compute $\lambda_t$. Please note that directly applying $\frac{\lambda_t}{2}\|w_t\|_2^2$ will invoke computation of 2nd order derivatives, since $\lambda_t$ is computed using the 1st order derivatives. Consequently, to save up computation and avoid 2nd order derivatives, we convert the $\lambda_t$ into a non-tensor scalar before plugging its value into the weight decay regularization term.

Using our approach, the value of $\lambda_t$ varies in each iteration. However, one could make $\lambda_t$ more stable, by using an exponential weighted average $\bar{\lambda}_t = 0.1 \times \bar{\lambda}_{t-1} + 0.9 \times \lambda_t$.

Algorithm 1 provides a psudo-code of our implementation.

## C    TUNING WD AND LR

As discussed previously, tuning the hyper-parameters such as $\lambda_{wd}$ is crucial. First, through examples, we demonstrate why the correct way of finding the right hyper-parameter for $\lambda_{wd}$ and $lr$ (learning rate) is a 2D grid search over different values of these hyper-parameters. For the sake of saving in computation, one might assume that fixing $lr$ and 1D grid search over $\lambda_{wd}$, then fixing $\lambda_{wd}$ and grid search over $lr$ might be a (computationally) cheaper and as effective.

---

**Algorithm 1** Adaptive Weight Decay

---

**Require:** $DoG > 0$
  $\bar{\lambda} \leftarrow 0$
  **for** $(x, y) \in loader$ **do**
    $p \leftarrow model(x)$                                           $\triangleright$ Get models prediction.
    $main \leftarrow CrossEntropy(p, y)$                          $\triangleright$ Compute CrossEntropy.
    $\nabla w \leftarrow backward(main)$          $\triangleright$ Compute the gradients of main loss w.r.t weights.
    $\lambda \leftarrow \frac{\|\nabla w\| DoG}{\|w\|}$            $\triangleright$ Compute iteration's weight decay hyperparameter.
    $\bar{\lambda} \leftarrow 0.1 \times \bar{\lambda} + 0.9 \times stop\_gradient(\lambda)$      $\triangleright$ Compute the weighted average as a scalar.
    $w \leftarrow w - lr(\nabla w + \bar{\lambda} \times w)$               $\triangleright$ Update Network's parameters.
  **end for**

---

Consider the following example (Figure 7) for training image classification networks on CIFAR-10 using Wide Residual Nets (WRN28-10) [4]. We have initial guesses $\lambda_{wd} = 0.005$ and $lr = 0.01$ [5]. We perform a 1D grid search for $lr$ first while fixing $\lambda_{wd} = 0.005$ (Figure 7(a)). Then after realizing that $lr = 0.01$ is the optimal hyper-parameter value, we perform a 1D grid search on $\lambda_{wd}$ (Figure 7(b)). No matter how many times we alternate between searching for $\lambda_{wd}$ and $lr$, we always will end up with the same fixed values of $\lambda_{wd} = 0.005$ and $lr = 0.01$. One might assume that these values are optimal, while $\lambda_{wd} = 0.0005$ and $lr = 0.1$ yields better results (Figure 7(c)).

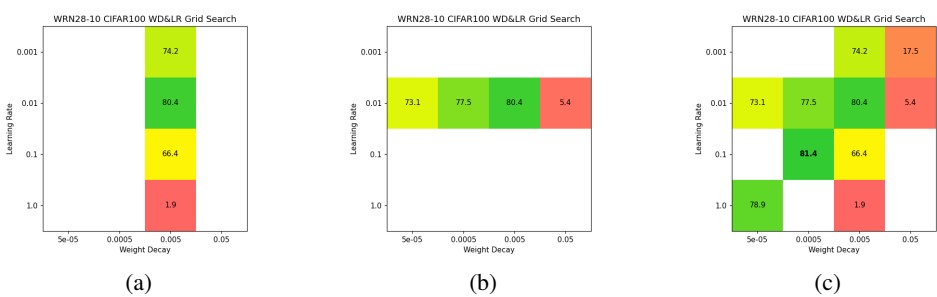

|     (a)     |     (b)     |     (c)     |

Figure 7: Held-out validation accuracy for different searches on tunable hyper-parameter values. Separate grid search on learning rate (a) and weight decay's hyper-parameter (b) does not reveal the optimal set of hyper-parameters, at the best it reveals the diagon on which the best hyper-parameters resides. An extra search on the diagon is necessary to find the said hyper-parameter (c).

## D   TUNING ADAPTIVE WEIGHT DECAY

Since, as explained before, adaptive weight decay is more robust to change in $lr$, it would be easier first to tune $DoG$ and then tune $lr$. Also, tuning $DoG$ and $lr$ for adaptive weight decay should be easier than tuning $\lambda_{wd}$ and $lr$ for non-adaptive weight decay.

While the preferred method for finding the best hyper-parameter for adaptive weight decay (i.e., $DoG$) is to treat it as any other hyper-parameter and perform a random grid-search, we present a method to find a good estimate which works well in practice. We suggest that if a training pipeline exists that is fully tuned for non-adaptive weight decay, we can get a reasonable estimate for $DoG$ by following steps mentioned in Fig. 2.

## E   EXTRA RESULTS

Here, we provide the extra results and figures not included in the body of the paper.

---

[4]details of experiments can be found in Appendix A
[5]a common choice for hyper-parameter values for training CIFAR resnets to build baselines

### E.1 CIFAR-10 ROBUSTNESS TO NOISY LABELS

The results of the experiments for training classifiers on the CIFAR-10 dataset with noisy labels can be seen in Fig. 8 which yields similar conclusions to that of CIFAR-100.

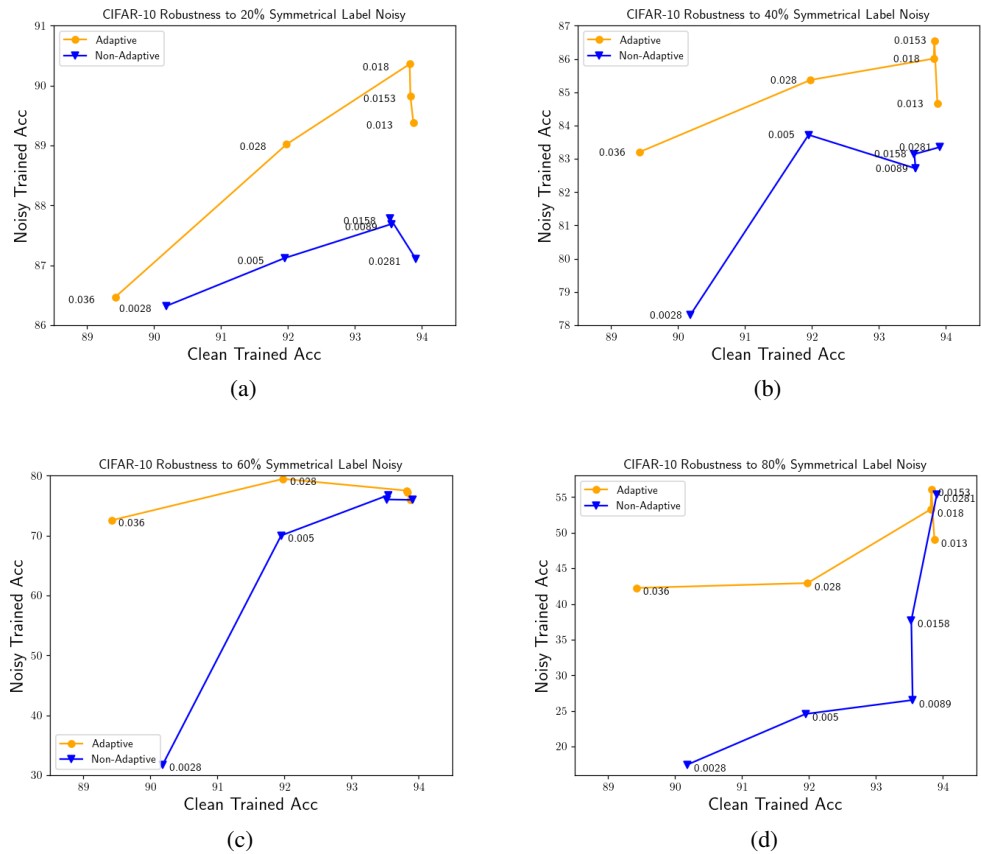

Figure 8: Comparison of similarly performing networks once trained on CIFAR-100 clean data, after training on 20% (a), 40% (b), 60% (c), and 80% (d) noisy data. Networks trained with adaptive weight decay outperform non-adaptive trained networks.

### E.2 2D GRID SEARCH FOR BEST PARAMETER VALUES FOR RESNET32

The importance of the 2D grid search on learning-rate and weight decay hyper-parameters are not network dependent. And we can see how these values are tied together for ResNet32 in Figure 9.

### E.3 VISUALIZING IMAGES FROM THE CIFAR-100 TRAINING SET WHERE BEST AWD MODELS DO NOT FIT.

In Fig. 10, we visualize some of the 4.71% examples which belong to the CIFAR-100 training set that our AWD trained network doese not fit. Interestingly enough, there are many examples like 0-3, 10-14, 48, and 49 with overlapping classes with one object. There are many examples with wrong labels, such as 4-6, 43-45, 28-29, and 51-57. In many more examples, there are at least two objects in one image, such as 15-25, 28-33, 38-42, and 58-59. Unsurprisingly, the model would be better off not fitting such data, as it would only confuse the model to fit the data, which contradicts its already existing and correct conception of the object.

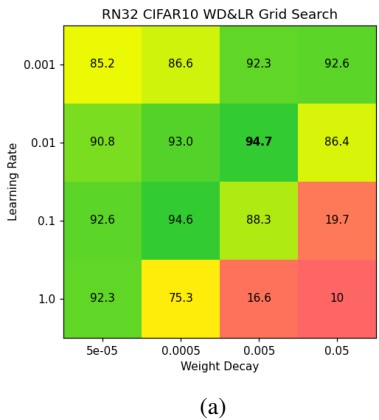 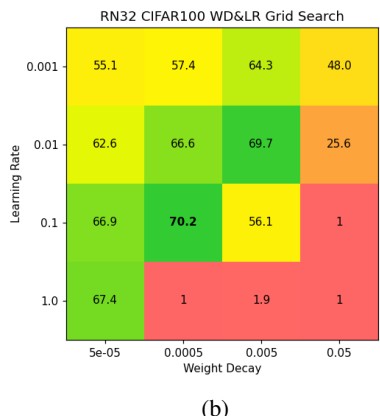

(a)                                                    (b)

Figure 9: Grid Search on different values of learning rate and weight decay on accuracy of ResNet32 on (a) CIFAR10 and (b) CIFAR100.

### E.4    ROBUSTNESS TO SUB-OPTIMAL LEARNING RATE

In this section, we have a deeper look the performance of networks trained with sub-optimal learning rates. We observe that adaptive weight decay is more robust to changes in the learning rate in comparison to non-adaptive weight decay for both CIFAR-10 and CIFAR-100, as shown in Figure 11(a) and 11(b). The accuracy for $lr = 1.0$ for non-adaptive weight decay drops 69.67% on CIFAR-100 and 5.0% on CIFAR-10, compared to its adaptive weight decay counterparts.

The robustness to sub-optimal learning rates suggests that adaptive weight decay might be more suitable for applications where tuning for the optimal learning rate might be expensive or impractical. An example would be large language models such as GPT-3 or Megatron-Turing NLG, where even training the network once is expensive (Brown et al., 2020; Smith et al., 2022; Rasley et al., 2020). Another example would be neural architecture search, where one trains many networks (Tan & Le, 2019; Zhou et al., 2018; Real et al., 2019; Bergstra et al., 2013; Mendoza et al., 2016).

### E.5    ADAPTIVE WEIGHT-DECAY TO IMAGENET

In this section we illustrate that tuning adaptive weight decay can result in comparable performance to non-adaptive weight-decay even in settings where training with non-adaptive weight decay does not suffer from over-fitting. For this purpose we perform free adversarial training (Shafahi et al., 2019) with $\epsilon = 4/255$ on ImageNet scale. We use a resnet-50 backbone. The parameters used in this setting are replay $m = 4$, batchsize of 512, and an initial learning rate of 0.1 which drops by a factor of 0.1 each $n/3$ epochs.

Similar to the experiments in the main body, we perform a hyper-parameter search to find the best weight-decay parameter and the best adaptive weight decay parameter and report both the robustness and clean accuracy of each of the trained models. The results are summarized in Table 6.

### E.6    ROBUSTNESS TO PARAMETER PRUNING

As seen in previous sections, models trained with adaptive weight decay tend to have smaller weight norms. Smaller weight norms, can indicate that models trained with adaptive weight decay are less sensitive to parameter pruning. We adopt a simple non-iterative $\ell_1$ pruning to test our intuition. Table 7 which shows the accuracy of models trained with adaptive and non adaptive weight decay after various percentages of the parameters have been pruned verifies our hypothesis. Adaptive weight decay trained models which are trained with various learning rates are more robust to pruning in comparison to models trained with non-adaptive weight decay.

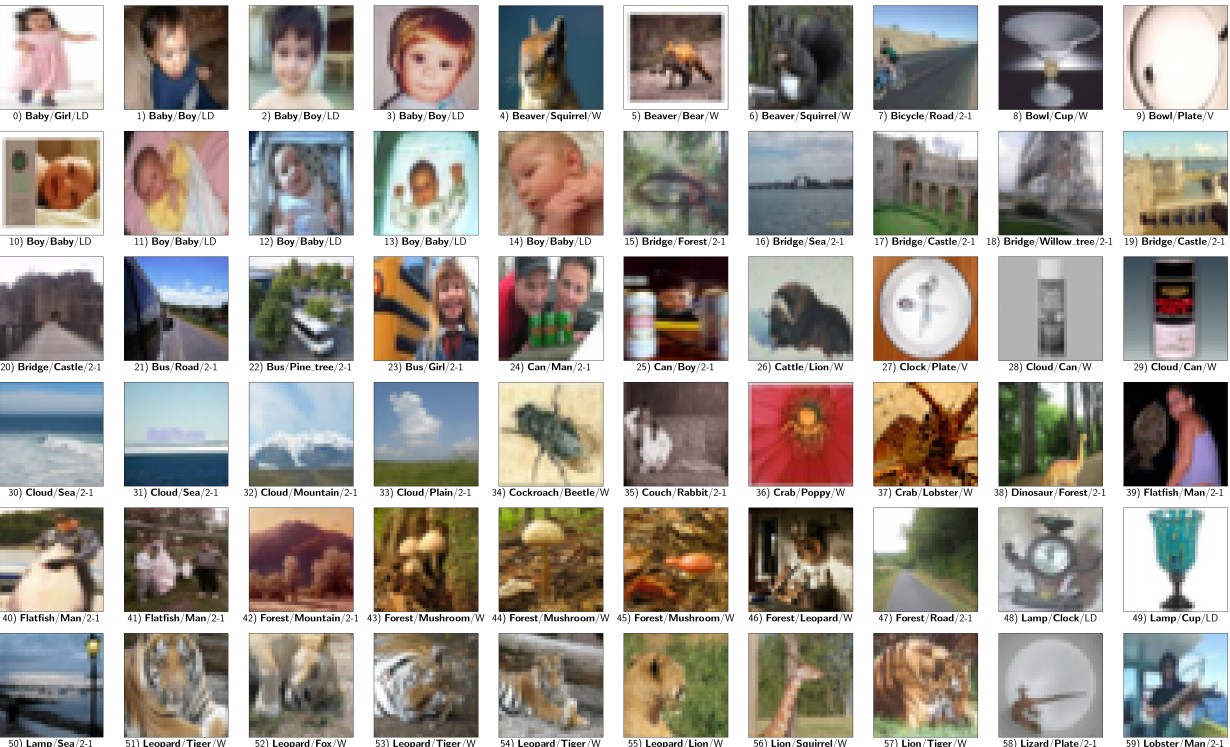

Figure 10: Examples from CIFAR-100 training dataset that have noisy labels. For every image we state [dataset label/ prediction of classifier trained with AWD / our category of noisy case]. We classify these noisy labels into several categories: **W**: Wrong Labels where the picture is clear enough to comprehend the correct label.; **2-1**: Two Objects from CIFAR-100 in one image, but only one label is given in the dataset; **LD**: Loosely Defined Classes where there is one object, but one object could be two classes at the same time. For instance, a baby girl is both a baby and a girl. **V**: Vague images where authors had a hard time identifying.

## F    RELATED WORK ON ADAPTIVE WEIGHT DECAY

The concept of Adaptive Weight Decay was first introduced by (Nakamura & Hong, 2019). Similar to our method, their method (AdaDecay) changes the weight decay's hyper-parameter at every iteration. Unlike our method, in one iteration, AdaDecay imposes a different penalty to each individual parameter, while our method penalizes all parameters with the same magnitude. More specifically, the updates from weight decay in our method is $-\lambda_t w$, where $\lambda_t$ varies at every iteration. However, the updates in AdaDecay for parameter $w_i$ is $-\lambda \theta_{t,i} w_i$, where $\lambda$ is constant at every iteration, instead, $0 \leq \theta_{t,i} \leq 2$ can vary for different parameters $i$ and for different iterations $t$. In other words, AdaDecay introduces $\theta_{t,i}$ for every single parameter of the network to represent how strongly each parameter should be penalized in the weight decay. For instance, if $\theta_{t,i} = 1$, then AdaDecay has the same effect as non-adaptive weight decay. For parameters in layer $L$, $\mathbb{E}_{w_i \in L} \theta_{t,i} = 1$, due to the fact that AdaDecay computes $\theta_{t,i}$ based on layerwise-normalized gradients. More percisely:

$$\theta_{t,i} = \frac{2}{1 + exp(-\alpha \nabla \bar{w}_{i,t})}$$

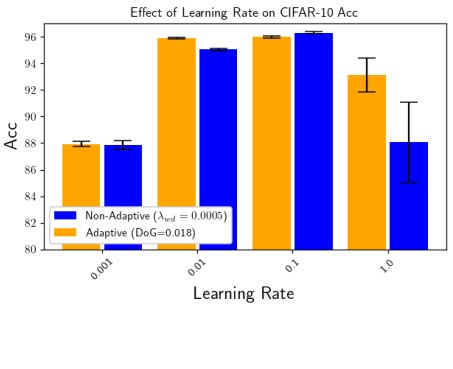 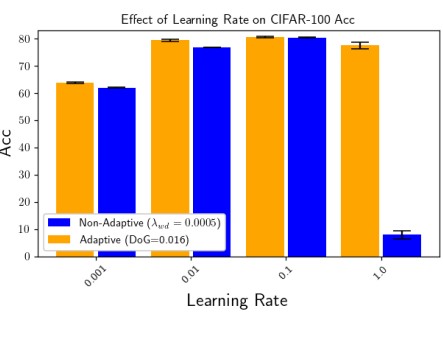

|(a)|(b)|

Figure 11: Models trained with Adaptive Weight Decay are more robust to Learning rate. Results are an average over 5 trials. We use the same experiment setup and estimated $DoG$ values from section 3.

| Method | Model | $\epsilon$ | robustness % | natural accuracy % |
|---|---|---|---|---|
| $\lambda_{wd} = 0.000004$ | Resnet-50 | 4 | 26.042 | 55.406 |
| $\lambda_{wd} = 0.000005$ | Resnet-50 | 4 | 26.706 | 54.946 |
| $\lambda_{wd} = 0.000006$ | Resnet-50 | 4 | 25.840 | 55.128 |
| $\lambda_{wd} = 0.000007$ | Resnet-50 | 4 | 25.706 | 54.202 |
| $DOG = 0.0006$ | Resnet-50 | 4 | 25.940 | 54.062 |
| $DOG = 0.0007$ | Resnet-50 | 4 | **26.844** | **56.310** |
| $DOG = 0.0008$ | Resnet-50 | 4 | 26.356 | 54.992 |
| $DOG = 0.0009$ | Resnet-50 | 4 | 26.390 | 55.694 |

Table 6: Robustness and clean accuracy of resnet-50 models trained with adversarial training for free $m = 4$ to be robust against attacks with robustness budget of $\epsilon = 4$.

where $\nabla \bar{w}_{i,t}$ is the layerwise-normalize gradients. More percisely, $\nabla \bar{w}_{i,t} = \frac{\nabla w_{i,t} - \mu_L}{\sigma_L}$ where $\mu_L$ and $\sigma_L$ represent mean and standard deviation of gradients at layer $L$.

The main difference between our version and AdaDecay apart from our different formulations is that our method can increase the weight norm penalty (i.e. $\lambda_t$) indefinitely, while AdaDecay on average penalizes the parameters of the same layer similar to non-adaptive weight decay. This is due to the fact that $\mathbb{E}_{w_i \in L} \theta_{t,i} = 1$. For instance, for the most extreme case, assuming that $\theta_{t,i} = 2$ for all $i$ and $t$, the effect of AdaDecay becomes at most twice as strong as non-adaptive weight decay, while our version does not have such upperbounds.

| Method | Learning Rate | | | |
|---|---|---|---|---|
| | 0.001 | 0.01 | 0.1 | 1.0 |
| Adaptive $DoG = 0.016$ | **92.53** | **16.22** | **18.3** | 26.65 |
| Non-Adaptive $\lambda_{wd} = 0.0005$ | 116.98 | 27.01 | 24.06 | **12.16** |

Figure 12: Norm of the weights for networks trained with adaptive weight decay and non-adaptive weight decay.

| Dataset | LR | Method | Nat | 40% | 50% | 60% | 70% | 80% | 90% |
|---|---|---|---|---|---|---|---|---|---|
| C10 | 1 | $DoG = 0.018$ | **93.1 ± 1.3** | **93.1 ± 1.3** | **93.1 ± 1.3** | **93.1 ± 1.3** | **93.1 ± 1.3** | **93.1 ± 1.3** | **93.2 ± 1.3** |
| | | $\lambda_{wd} = 0.0005$ | 88.1 ± 3.0 | 88.0 ± 3.1 | 88.0 ± 3.1 | 88.0 ± 3.1 | 88.0 ± 3.1 | 88.0 ± 3.1 | 88.0 ± 3.1 |
| | 0.1 | $DoG = 0.018$ | 96.0 ± 0.1 | 91.6 ± 8.7 | 91.7 ± 8.6 | 91.7 ± 8.6 | 91.5 ± 9.0 | 90.5 ± 10.5 | 83.7 ± 17.9 |
| | | $\lambda_{wd} = 0.0005$ | **96.3 ± 0.1** | 93.6 ± 5.3 | 93.6 ± 5.3 | 93.6 ± 5.3 | 93.4 ± 5.5 | 92.6 ± 6.2 | 84.3 ± 13.6 |
| | 0.01 | $DoG = 0.018$ | **95.9 ± 0.1** | **95.9 ± 0.1** | **95.9 ± 0.1** | **95.9 ± 0.1** | **95.9 ± 0.1** | **95.8 ± 0.1** | **91.4 ± 1.4** |
| | | $\lambda_{wd} = 0.0005$ | 95.0 ± 0.1 | 95.0 ± 0.1 | 95.0 ± 0.1 | 94.5 ± 0.3 | 75.2 ± 12.9 | 19.3 ± 12.6 | 12.2 ± 5.0 |
| | 0.001 | $DoG = 0.018$ | 87.9 ± 0.2 | 82.1 ± 1.9 | 75.2 ± 4.2 | 64.6 ± 7.2 | 38.3 ± 8.4 | 18.8 ± 4.3 | 13.4 ± 1.9 |
| | | $\lambda_{wd} = 0.0005$ | 87.9 ± 0.4 | 80.5 ± 2.9 | 73.8 ± 4.8 | 56.4 ± 8.1 | 46.8 ± 16.9 | 21.0 ± 9.9 | 12.2 ± 2.5 |
| C100 | 1 | $DoG = 0.016$ | **77.6 ± 1.3** | **77.6 ± 1.3** | **77.6 ± 1.3** | **77.6 ± 1.3** | **77.6 ± 1.3** | **77.5 ± 1.2** | **75.3 ± 1.3** |
| | | $\lambda_{wd} = 0.0005$ | 7.9 ± 1.4 | 2.6 ± 1.8 | 2.6 ± 1.8 | 2.6 ± 1.8 | 2.6 ± 1.8 | 2.6 ± 1.8 | 2.6 ± 1.8 |
| | 0.1 | $DoG = 0.016$ | 80.7 ± 0.3 | 80.7 ± 0.3 | **80.7 ± 0.3** | **80.5 ± 0.2** | **79.6 ± 0.4** | **74.5 ± 0.9** | **29.2 ± 3.6** |
| | | $\lambda_{wd} = 0.0005$ | 80.5 ± 0.2 | 80.4 ± 0.1 | 80.2 ± 0.2 | 79.9 ± 0.2 | 78.4 ± 0.2 | 71.3 ± 1.2 | 23.9 ± 6.7 |
| | 0.01 | $DoG = 0.016$ | **79.5 ± 0.4** | **79.5 ± 0.3** | **79.4 ± 0.3** | **79.3 ± 0.4** | **78.7 ± 0.5** | **75.0 ± 0.8** | **28.9 ± 4.4** |
| | | $\lambda_{wd} = 0.0005$ | 76.9 ± 0.1 | 74.4 ± 3.6 | 74.1 ± 3.8 | 73.5 ± 4.2 | 67.2 ± 4.1 | 17.5 ± 6.9 | 1.5 ± 0.4 |
| | 0.001 | $DoG = 0.016$ | **63.8 ± 0.3** | **55.3 ± 1.8** | **47.4 ± 2.2** | **32.0 ± 2.1** | 12.9 ± 2.9 | 2.8 ± 1.4 | 1.1 ± 0.2 |
| | | $\lambda_{wd} = 0.0005$ | 62.1 ± 0.2 | 52.7 ± 2.0 | 41.7 ± 2.7 | 25.6 ± 2.4 | 11.0 ± 4.4 | 3.1 ± 0.9 | 1.2 ± 0.1 |

Table 7: Models trained with Adaptive Weight Decay are more robust to change in learning rate and pruning.

