# OpenReview forum: "Adaptive Weight Decay: On The Fly Weight Decay Tuning for Improving Robustness"
_ICLR.cc/2023/Conference — Submitted to ICLR 2023_

### Official Review · Reviewer_4bPN · 2022-10-24

**Confidence:** 5
**Correctness:** 3
**Technical Novelty And Significance:** 3
**Empirical Novelty And Significance:** 3
**Recommendation:** 6

**Clarity, Quality, Novelty And Reproducibility:**

The clarity and quality are good. The proposed method is simple, but its novelty is not a major concern to me.

**Strength And Weaknesses:**

Strength:

The paper is clearly written and the experiment on CIFAR10/100 is extensive. The paper evaluates the AWD not only in the clean training setting, but in adversarial and noisy label setting, which all show the effectiveness of AWD over weight decay. I kind of like the style of this paper: proposing a simple but clever method and showing its effectiveness in a wide range of scenarios.


Weaknesses:

1. The first concern is about the hyperparameter in the AWD regularization. The DoG=0.016 is determined based on empirical results on CIFAR100 and used in other settings (only CIFAR10) in this paper. I wonder if the DoG can be determined by simple hyperparameter tuning like grid search. If the hyperparameter can only be determined by first training with weight decay, then I cannot say the AWD is a general regularization. I would recomment to show the performance of AWD with a simple hyperparameter tuning on other tasks such as object detectioc, sementic segmentation or at least other image classification datasets such as ImageNet to demonstrate the its general effectiveness.

2. My second concern is about the motivation. The AWD is proposed to keep the ratio between gradient and weight panelty in the SGD update. But the AWD does not have the same SGD update form as in Equation (1), since the gradient norm is also backpropagated if my understanding is correct (based on the discussion following Equation (5)). In other words, if the motivation is to keep the DoG, one should compute the $\lambda_{WD}$ based on the weight norm and its gradient norm and keep the original weight decay regularization. Please correct me if you do not take the gradient of the gradient norm, since there is no code as a reference.

Minor Issue:

1. The proposed method is not compared with other adversarial defense method, but the evaluation is kind of standard so it is easy to have a comparison. I suggest adding some baselines as a reference to better position the proposed method.

2. The regularization approach to adversarial robustness is not fully discussed. For example, the effective margin regularization [1] penalizes the weight gradient norm to boost the adversarial robustness, which looks similar to this paper. Could the author provide a discussion on the diffierence and connection?

[1] Ziquan Liu and Antoni B. Chan, “Boosting Adversarial Robustness From The Perspective of Effective Margin Regularization.” British Machine Vision Conference (BMVC), 2022

**Summary Of The Paper:**

The paper proposes to use an adaptive weight decay (AWD) as a regularization, inspired by the observation that the ratio between gradient and weight in weight decay training is a more fundamental parameter than $\lambda_{WD}$. The paper shows that AWD has a benefit in clean training, adversarial training and learning with noisy labels with rigourous experiment evaluation. Finally, AWD is shown to alleviate the overfitting phenomenon in CIFAR10 since the misclassified samples are mostly mis-labeled.

**Summary Of The Review:**

Overall, I would like to give a borderline score at this phase and I am happy to increase it if my concerns can be addressed.

---

> ### Author Response · Authors · 2022-11-16
> **Thank you for your feedback, Reviewer 4bPN  !**
>
> Thank you for your constructive and insightful comments! Please see our responses below.
>
> >*I wonder if the DoG can be determined by simple hyperparameter tuning like grid search.*
>
> Yes, you are absolutely correct. We actually do simple hyperparameter tuning grid search like you suggested for the experiments in Figure 3 and Figure 4.
> In fact, the adaptive weight decay does not eliminate the need for the hyper-parameter search for weight decay. Instead, it introduces a more effective weight decay method that still benefits from hyper-parameter tuning.
>
> >*I would recommend to show the performance of AWD with a simple hyperparameter tuning on other tasks such as object detection, semantic segmentation or at least other image classification datasets such as ImageNet to demonstrate the its general effectiveness.*
>
> Per your request, we have included new robustness experiments similar to Figure 3, on ImageNet to test our method at large scale. The results can be found in the general authors’ response.
>
>
> >*Please correct me if you do not take the gradient of the gradient norm, since there is no code as a reference.*
>
> We apologize for the confusion and thank you for this comment. As explained in the general authors’ response, we do not compute the hessians in our algorithm and instead treat the computed weight decay value as a scalar that does not propagate gradients.  Consequently, the difference in the running time between the adaptive weight decay and the non-adaptive version is negligible.

---

> > ### Comment · Reviewer_4bPN · 2022-12-02
> > **Followup comment**
> >
> > Thanks for the response. My major concern is addressed after reading the response and I raise my score to 6.

---

### Official Review · Reviewer_d8cT · 2022-10-24

**Confidence:** 3
**Correctness:** 3
**Technical Novelty And Significance:** 2
**Empirical Novelty And Significance:** 2
**Recommendation:** 3

**Clarity, Quality, Novelty And Reproducibility:**

For writing, most parts of this work are well written and clear.

For novelty, the novelty of this work is not high since its only contribution is to use the ratio of gradient norms of the loss and the regularization as a metric to trade-off the updating. Moreover, this method needs to compute Hessian which is very expensive and limits the method to use in large networks.


**Strength And Weaknesses:**

Strengths:
There are main two contributions in this work.
1)The proposed  method is quite simple.


2)The experimental improvement is notable compared with the vanilla regularization.


Weaknesses:
1)The proposed method is slightly incremental, and brings extra computational cost. The only contribution is that it uses the ratio of gradient norms of the loss and the regularization as a metric to trade-off the updating, which is not novel. Moreover, the gradient norm occurs in the loss, which means that it needs to further compute hessian matrix for backpropagation. But in practice, it is really hard to compute Hessian matrix, and even be inhibitively expensive to compute for large networks.  So this method cannot be used in the modern networks.


2)In the experimental section, the authors only compare few baselines which cannot actually reflect the superiority of the proposed method. Moreover,  the experiments also lack of large experiments, such as some experiments on ImageNet which is a standard dataset to test the performance nowadays.  Finally, as discussed above, this method needs to compute Hessian which is very expensive. So it is better to compare the training time of all baselines.



**Summary Of The Paper:**

In this work, the authors propose adaptive weight decay.  This strategy replaces the hyper-parameter of weight decay by the product of a hyper-parameter and the ratio of gradient norm and parameter norm. In this way, it can automatically tune the hyperparameter for weight decay during each training iteration. Then the authors show the effectiveness of the proposed method on adversarial robustness.

**Summary Of The Review:**

Overall, this work provides some good empirical results. But it fails to provide good new insights and practical techniques.

---

> ### Author Response · Authors · 2022-11-16
> **Thank you for your feedback, Reviewer d8cT !**
>
> Thank you for your questions and comments! Please see our responses below.
>
> >*Moreover, the gradient norm occurs in the loss, which means that it needs to further compute hessian matrix for backpropagation.*
>
> As discussed in the General authors’ response, we do not compute hessians in our algorithm, due to the fact that we treat the weight decay hyper-parameter in the adaptive weight decay as a scalar number and we stop the gradients from backpropagating through this variable. Consequently, the difference in the running time between the adaptive weight decay and the non-adaptive version is negligible.
>
>
> >*Moreover, the experiments also lack of large experiments, such as some experiments on ImageNet which is a standard dataset to test the performance nowadays.*
>
> To illustrate the breadth and various benefits of our approach in different settings such as noisy labels, adversarial training, and pruning, we were forced to limit the computation and run extensive analysis on CIFAR-10 & CIFAR-100 which are common datasets for evaluating robustness. However, per your request and the request of other reviewers, we added new robustness experiments similar to Figure 3, on ImageNet. The summary can be found in the general authors’ response.
>
>
> >*In the experimental section, the authors only compare few baselines which cannot actually reflect the superiority of the proposed method…So it is better to compare the training time of all baselines.*
>
> As mentioned in our responses above, adaptive weight decay algorithm does not compute hessians. In fact both algorithms have the same number of forward and backward passes through the network and have a similar running time. Also, note that using AdaptiveSAM requires twice as many forward/backward passes, so in order to compensate for the extra cost of using AdaptiveSAM, the  rows that use AdaptiveSAM for their optimization are optimizing for half of the number of epochs compared to other experiments.
> This is in fact one of the strengths of our method. Note that all rows in Table 4 have the same running time with the exception of the last two rows. Please note that we achieve 29.54% robustness accuracy with almost half number of steps, compared to Rebufi et. al. which achieves 29.80% with twice as many iterations and more complicated augmentations.

---

### Official Review · Reviewer_N7iq · 2022-10-25

**Confidence:** 4
**Correctness:** 2
**Technical Novelty And Significance:** 2
**Empirical Novelty And Significance:** 2
**Recommendation:** 5

**Clarity, Quality, Novelty And Reproducibility:**

This paper is well organized. However, some claims or sentences are confusing (I pointed out some in the “Weaknesses” section).  The empirical contribution may be significant according to the adversarial robustness gain. The authors provide the training details and pseudocode of adaptive weight decay, which provides good reproducibility.

**Strength And Weaknesses:**

Strength
+ The motivation for dynamically finetuning weight decay is clear. By showing the results by grid search, it is difficult and also important for us to find an optimal weight decay during training.
+ The empirical results support the claim that adaptive weight decay can improve robustness.

Weaknesses
- I am a bit confused about the functionality of the value of DoG at the epoch when the loss is at the plateau in Section 2. The DoG value is found based on a specific value of the weight decay hyperparameter. Why does keeping the value of DoG constantly in every optimization help training if the DoG is the significant underlying difference between the performance of different cells in the same diagonal? Also, I am confused about the meaning of “significant underlying difference” and the reason for that “significant underlying difference” can be evaluated by DoG.
- Although the authors provide some intuitive reasons for why keeping DoG constantly can help training, it still lacks some theoretical analyses to make the claim solid enough.
- I am concerned that whether the results of AutoAttack in Table 3 are obtained at the last epoch. It would be better to provide and compare the best AutoAttack accuracy that is selected based on a validation set between the proposed method and the baseline.

Minor comments
+ The label of the blue line in Figure 2(b) should be “Log(loss)”?
+ It seems that the loss plateaus is at Epoch 400 in Figure 2(b)? The loss value at Epoch 350 is higher than that at Epoch 400.



**Summary Of The Paper:**

This paper proposes a dynamic scheduling strategy for the value of the weight decay hyperparameter. During the training procedure, the authors keep the ratio of the value of weight decay value to the gradient of cross-entropy loss (dubbed as DoG) at a constant. Empirically, this adaptive weight decay strategy can help improve the adversarial robustness and gain a smaller drop in performance in the noisy-label setting.

**Summary Of The Review:**

This paper proposes to adaptively finetune the value of weight decay hyperparameter according to DoG. Although the empirical results could be significant, I have several concerns that are stated in the “Weaknesses” section.

---

> ### Author Response · Authors · 2022-11-16
> **Thank you for your feedback, Reviewer N7iq !**
>
> Thank you for your questions and comments! Please see our responses below.
>
> >*I am concerned that whether the results of AutoAttack in Table 3 are obtained at the last epoch. It would be better to provide and compare the best AutoAttack accuracy that is selected based on a validation set between the proposed method and the baseline.*
>
> We agree that this approach for early-stopping is the correct approach and is indeed the approach taken in the paper. As we mention in the paper on page 5: “For early-stopping, we select the checkpoint which gives the highest $\ell_\infty = 8$ robustness accuracy measured by a 20 step PGD attack on the held-out validation set”.
> So all adversarial robustness tables and figures are evaluated using AutoAttack on the highest performing checkpoints on a held-out validation set.
>
>
>
> >*The label of the blue line in Figure 2(b) should be “Log(loss)”?*
>
> We will correct this mistake in the updated manuscript. Thank you for sharing that with us.
>
> >*I am a bit confused about the functionality of the value of DoG at the epoch when the loss is at the plateau in Section 2.*
>
> After the loss plateaus, the updates to weights become smaller and smaller and the generalizability of the network does not change. We wanted to study the average DoG for iterations that actually change the performance (i.e., there still is a competition between the cross-entropy loss and the weight-decay term) and mimic that behavior using the adaptive version. Because of that, it made sense to track when the loss plateaus.
>
>
> >*Why does keeping the value of DoG constantly in every optimization help training if the DoG is the significant underlying difference between the performance of different cells in the same diagonal*
>
> Thanks for asking this. Our intuition is that achieving this amount of DoG during training is what helps generalization. One way to achieve this, is to use a fixed hyper-parameter for non-adaptive weight decay that results in this value of DoG (i.e. best hyper-parameter for non-adaptive version).. Another way is to keep DoG constant at every iteration (i.e. our adaptive version). This is just a choice we made for simplicity. We did several other experiments which we did not include in the paper that achieved similar average DoG and performed similarly. For instance, instead of keeping DoG constant, if we start the DoG at zero in the first epoch and linearly increase it to 2xDoG until the last epoch (so that it would result in average DoG of 0.016), we saw a similar performance.
>
>
> >*It seems that the loss plateaus is at Epoch 400 in Figure 2(b)? The loss value at Epoch 350 is higher than that at Epoch 400.*
>
> Note that the plot is drawn in the logarithmic scale, so large differences at the end of training means very small changes in very small magnitudes. For instance, the loss at Epoch 350 is ~0.0060 compared to loss at Epoch 400 which is ~0.0056.

---

### Official Review · Reviewer_ksvx · 2022-10-27

**Confidence:** 3
**Clarity, Quality, Novelty And Reproducibility:** Well reproducible. See above section …
**Correctness:** 3
**Technical Novelty And Significance:** 3
**Empirical Novelty And Significance:** 2
**Recommendation:** 5

**Strength And Weaknesses:**

Strengths:
1) The problem that is solved by AWD is real and is well demonstrated by the initial experimentation of CIFAR with 2D grid search.
2) The idea of AWD is extremely clean and empirically informed.
3) The initial experimentation to show the value of AWD is clean.
4) The explanation of potential use cases of AWD along with adversarial robustness is very helpful for a new reader.
5) The gains on "robust accuracy" with adversarial training compared to vanilla WD and other baselines are empirically strong.
6) Rest of the analysis is a good step toward verifying the use of AWD

Weaknesses:
1) The paper at times is hard to parse.
2) While AWD is empirically informed, the initial choice of 0.016 opens more avenues of grid search as shown in the rest of the experiments.
3) The authors have considered a very specific setting where the learning rate doesn't decay. I am not sure if the learning rate decay is a huge factor as it often is for ImageNet scale training.
4) While "adversarial training" is an interesting thing to try out in the context of a white box attack, it often is not the robustness we care about. We care about OOD robustness akin to "do cifar-10 classifiers generalize to cifar-10?" and "do imagenet classifiers generalize to imagenet?". I think this would be one of my biggest questions if AWD makes things robust in general or is it due to the fact that AWD has a notion of running gradient that approximates to hessian? Because the use of hessian approximations for optimizations could assist in adversarial robustness when trained for PGD attacks. I would like the author's thoughts on this.
5) The figures and tables of the paper are extremely hard to understand and are often not accessible for ease of reading. It would be great to fix that for a better dissemination of results and motivation.
6) Lastly, probably my biggest qualm is the lack of results at scale. While CIFAR-10/100 are great datasets to prototype on, it is often rare to see full generalization to ImageNet -- I would like to see at least one comparison at the ImageNet scale -- even if not for the adversiral robustness.

With all these factors in mind and considering my lack of expertise -- I am happy to chat with the authors during rebuttal and with revisions.  And consider increasing the score after the overall process. While the idea is simple, that does not take away novelty from it. However, the ad-hoc choice of factors like DoG does not sit well without some explanation or understanding.


**Summary Of The Paper:**

This paper proposed a simple technique to adaptively choose the weight decay parameter over the course of training, instead of having a fixed $\lambda$ which is currently the common practice, and call it adaptive weight decay (AWD). The motivation for AWD comes from the need to avoid a full expensive grid search of learning rates and weight decays. At the same time, the authors show that AWD helps with "adversarial training" and the downstream accuracy metrics.

The experimentation is on CIFAR-10 and 100 alongside baselines corresponding to adversarial training.

Note that I am not well versed in this sub-field of machine learning and I am genuinely confused about a lot of things in the paper with the exposition and experimentation. I will be providing the review from a general ML and scientific rigor perspective and would defer to other reviewers for expert opinions. Please excuse the brevity of the review owing to these reasons.

**Summary Of The Review:**

Simple and empirically driven idea to adaptively choose weight decay during the course of training. However, has issues with writing, presentation and some experimentation to be ready for being published.

---

> ### Author Response · Authors · 2022-11-16
> **Thank you for your feedback, Reviewer ksvx!**
>
> Thank you for your questions and comments! Please see our responses below,
>
> >*The authors have considered a very specific setting where the learning rate doesn't decay. I am not sure if the learning rate decay is a huge factor as it often is for ImageNet scale training.*
>
> We have included all the experimental details in the appendix and we realize that it has caused confusion. As explained in the Appendix A, the learning rate actually decays in ALL of the experiments with a cosine decay schedule starting at the initial learning rate and it decays to zero. In the newest experiments on ImageNet, we have used the conventional step-wise schedule learning rate decay.
>
>
>
>
> >*Lastly, probably my biggest qualm is the lack of results at scale. While CIFAR-10/100 are great datasets to prototype on, it is often rare to see full generalization to ImageNet -- I would like to see at least one comparison at the ImageNet scale -- even if not for the adversarial robustness.*
>
> Per your request, we did adversarial robustness experiments on the ImageNet dataset. The results have been included in the general authors’ response which illustrates the generalizability of the approach to larger datasets.
>
>
>
>
> >*While AWD is empirically informed, the initial choice of 0.016 opens more avenues of grid search as shown in the rest of the experiments.*
>
> You are absolutely correct. We do not suggest that adaptive weight decay eliminates the need for hyper-parameter search. Instead, we introduce a more effective weight decay algorithm that still benefits from hyper-parameter tuning.
>
>
>
>
> >*The figures and tables of the paper are extremely hard to understand and are often not accessible for ease of reading. It would be great to fix that for a better dissemination of results and motivation.*
>
> We thank the reviewer for this comment. We understand that the inclusion of many parts in the Appendix have caused the paper to be harder to follow. In the revision we will update the paper and include explanations to address the main confusion points mentioned by the reviewers.
>
>
>
>
>
> >*AWD makes things robust in general or is it due to the fact that AWD has a notion of running gradient that approximates to hessian?*
>
> As we discussed in the general authors’ response, we do not compute hessians. We attribute the improvement in all experiments to the fact that AWD is a more effective regularizer which maintains a balance between the different terms of the overall loss function. We believe that in adversarial robustness, it prevents robust overfitting more effectively and in the noisy-label and natural training settings, it prevents fitting on the noisy labeled data.

---

> > ### Comment · Reviewer_ksvx · 2022-11-19
> > **Thanks for the Rebuttal - Delayed Response**
> >
> > Thanks to the authors for their rebuttal. I will get back to this in a day or so, sorry for the delay.

---

> > ### Comment · Reviewer_ksvx · 2022-11-20
> > **Response to rebuttal**
> >
> > Dear Authors,
> >
> > Thanks for your rebuttal and updates. It is good to see the ImageNet experiments as well as the clarification with the implementation details.
> >
> > Coming to hessians, I have not mentioned you compute hessian, but implied a running stats of gradient often can approximate hessian. I think that is a fair point which still isn't fully thought about and I encourage you to that.
> >
> > I still don't see improvement in tables and figures, fixing them would help. Lastly, I have not heard back from authors on "robustness" experiments rather than adversarial robustness.
> >
> > I am happy to look at the responses when they come in, but as mentioned before I am not an expert in this field and would defer to the discussion between other reviewers for the core contributions.
> >
> > Thanks

---

### Author Response · Authors · 2022-11-16
**General Authors' Response**

We would like to thank all the reviewers for their constructive and insightful comments. Here, we address the most common comments that reviewers pointed out

Reviewers ksvx, d8ct and, 4bPN, asked for more experiments on larger datasets such as ImageNet. Per your request, we ran experiments on the ImageNet dataset. We decided to reproduce experiments similar to Figure 3 on ImageNet to show the effect of our method at a larger scale. Due to the limited time for rebuttal, we adopted the Adversarial Training for Free method with a batchsize of 512 and replay m = 4 for training robust models. We did a hyper-parameter search for both non-adaptive and adaptive weight decay hyper-parameters similar to Figure 3. Our results show that adaptive weight decay can scale to larger datasets – the best adaptive weight decay trained network **improves robustness by 0.14% while improving natural accuracy by 1.36%**. We associate this comparatively smaller gain (ImageNet vs CIFAR) in robustness to the fact that adversarial training on ImageNet (at least without huge number of epochs) does not suffer from [robust] overfitting. We verify this by illustrating that the best checkpoints are often the latest for ImageNet. Robust over-fitting is a concept which in the literature has been discussed extensively on CIFAR-100 and CIFAR-10, and most methods designed to target this phenomenon mainly show results on CIFAR-10. Nonetheless, we are glad that our results on ImageNet illustrate at least on-par performance to traditional weight-decay which in turn illustrates the broadness of our approach.


Both reviewers d8ct and 4bPN had concerns about computing hessians. We would like to clarify that in all experiments conducted in the paper **we have not computed second order derivatives**. The weight decay hyper-parameter $\lambda$ computed for adaptive weight decay depends on first order derivatives. After computing its value, we use it as a scalar floating point number that does not backpropagate the gradients. The adaptive and non-adaptive weight decay algorithms both require the same amount of computations for the forward and backward passes, with the exception of a negligible extra computation in the adaptive version for computing the norm of gradients and norm of weights in order to compute the weight decay hyper-parameter.
To keep the paper focused on the main results, we decided to include the implementation details and the pseudo-code of our algorithm in the Appendix. In hindsight, we realize that it might have not been our best decision and we apologize for the confusion. Based on Appendix B:
"Please note that directly applying $\frac{ \lambda_{t} }{2} || w_t ||^2$ will invoke computation of 2nd order derivatives, since $\lambda_t$ is computed using the 1st order derivatives.
Consequently, to save up computation and avoid 2nd order derivatives, we convert the $\lambda_{t}$ into a non-tensor scalar before plugging its value into the weight decay regularization term."

In the revised version, we will clarify this when first introducing AWD.

---

### Decision · Program_Chairs · 2023-01-20

**Decision:**

Reject

**Justification For Why Not Higher Score:**

The weaknesses mentioned in the general review:
* Is this about adversarial robustness, robustness in general, or general accuracy? Different evaluation, framing, and caveats will be required for each.
* This paper needs some larger-scale, standard training experiments either way to highlight the limits of this technique, and more if this is a general technique.
* I think this paper would really benefit from some theory about what this adaptive weight decay is really doing. I don't say that often, but weight decay is something about which we do have a fundamental understanding, so it merits it.

**Justification For Why Not Lower Score:**

N/A

**Metareview: Summary, Strengths And Weaknesses:**

**Summary:** This paper presents an adaptive weight decay (AWD) method where the weight decay value is scaled by the ratio of the gradient norm to the weight norm on each step of training. This method adds relatively little overhead but allows the strength of weight decay to change throughout training. Having some sort of adaptive weight decay is appealing. Weight decay is everywhere in deep learning, and we should probably be scheduling it similar to learning rate; it's just too expensive to do the joint exploration of LR, WD value, LR schedule, and WD schedule. Having an AWD scheme that eliminates the need to search for a WD schedule but gives benefits above and beyond constant WD would be very valuable to practitioners (including me). I was very excited about this possibility when I was reading the paper.

**Strengths:**
* This method is simple.
* This method seems to lead to meaningful improvements in adversarial robustness on its own on CIFAR tasks, without significant sacrifices in natural accuracy.
* This method also seems to be helpful for OOD robustness.

**Weaknesses:**
* The biggest concern reviewers raised (in my view) was how this method performed on natural training for larger-scale benchmarks like ResNet-50/ImageNet and an NLP task of choice (e.g., MLM or autoregressive language modeling). The authors updated the paper to include an adversarial robustness benchmark on ImageNet, but that did not directly satisfy this need. The reason this is so important is because the authors advertise this technique as a general replacement for weight decay. If it's really a general technique to replace weight decay, the authors _need_ to do their homework to make clear that this is a safe choice, i.e., that it isn't going to completely crash and burn in certain important settings that someone might want to use. The paper does not remotely meet that bar, focusing only on CIFAR. If the authors had advertised this technique as a simple way to improve adversarial robustness and only adversarial robustness, this might be closer to the bar for publication, but more is required given the more generic improvements to "robustness" used to sell the paper. I'm guessing that, had they sold the paper this way, they would have gotten different reviewers with more expertise on adversarial robustness to thoroughly evaluate whether this technique holds up under scrutiny in that community. And even if the paper were sold that way, I think reviewers would/should still be interested in seeing accuracy when doing standard training on large-scale tasks, if only to know the limits of this technique.
* This paper lacks any theoretical analysis of the change to weight decay. I'm not usually a fan of expecting theoretical analysis in papers and I typically think empiricism is enough. But weight decay is one of the few things we do in deep learning for which we do have a reasonable theoretical understanding. Given that the authors are tinkering with weight decay and provide only an intuition for what it might be doing, I think it's important to have some amount of theory in a paper like this, particularly if it is framed as a general contribution to the way we do regularization rather than a narrow trick for adversarial robustness.

**Recommendation:** I recommend rejection. This paper has an enormous amount of promise, and I urge the authors to submit again in the future. To have more success in the future, the authors need to commit to a story. If it's just about adversarial robustness and nothing else, commit to that and be as rigorous as possible in line with the standards of that community; if it's about a general replacement for weight decay, this paper needs to focus on larger-scale experiments. I personally think a minimal number of larger-scale, standard training experiments are needed either way, if only to highlight the limits of this method.

**Summary Of Ac-Reviewer Meeting:**

N/A